# The Influence of Surface Mycobiota on Sensory Properties of "Istarski pršut" and "Dalmatinski pršut"

**Tina Lešić** [1], **Nada Vahčić** [2], **Ivica Kos** [3], **Manuela Zadravec** [4], **Dragan Milićević** [5], **Irena Perković** [6], **Eddy Listeš** [7] and **Jelka Pleadin** [1,*]

1 Laboratory for Analytical Chemistry, Croatian Veterinary Institute, Savska Cesta 143, 10000 Zagreb, Croatia; lesic@veinst.hr
2 Faculty of Food Technology and Biotechnology, University of Zagreb, Pierottijeva 6, 10000 Zagreb, Croatia; nvahcic@pbf.hr
3 Department of Animal Science and Technology, Faculty of Agriculture, University of Zagreb, Svetošimunska Cesta 25, 10000 Zagreb, Croatia; ikos@agr.hr
4 Laboratory for Feed Microbiology, Croatian Veterinary Institute, Savska Cesta 143, 10000 Zagreb, Croatia; zadravec@veinst.hr
5 Institute of Meat Hygiene and Technology, Kaćanskog 13, 11040 Belgrade, Serbia; dragan.milicevic@inmes.rs
6 Regional Veterinary Institute Vinkovci, Croatian Veterinary Institute, Ul. Josipa Kozarca 24, 32100 Vinkovci, Croatia; perkovicirena1512@gmail.com
7 Regional Veterinary Institute Split, Croatian Veterinary Institute, Poljička Cesta 33, 21000 Split, Croatia; e.listes.vzs@veinst.hr
* Correspondence: pleadin@veinst.hr

**Abstract:** This study aimed to identify surface mould species overgrowing the Croatian protected meat products "Istarski pršut" and "Dalmatinski pršut" and their effect on sensory properties. Dry-cured hams were produced in 2018/2019 and obtained from annual fairs. The predominant surface species found on "Dalmatinski pršut" were *Aspergillus chevalieri*, *Penicillium citrinum* and *Aspergillus cibarius*, whereas those overgrowing "Istarski pršut" were *Aspergillus proliferans*, *P. citrinum* and *Penicillium salamii*. The results show species diversity, higher presence, and greater variety of *Aspergillus* species in "Dalmatinski pršut" in comparison to "Istarski pršut", and significant variations in 9 of 20 sensory attributes. Principal component analysis revealed a clear distinction between the two, and a large contribution of *P. salamii* and *Penicillium bialowienzense* to one principal component. The texture traits, smoky odour, muscle and subcutaneous fatty tissue colour, and mould species found are valuable for product characterisation. The results also indicate that mould species may be responsible for some sensory traits, such as tenderness, juiciness, and lesser freshness.

**Keywords:** meat products; dry-cured hams; sensory evaluation; surface moulds; *Penicillium*; *Aspergillus*; Croatian regions

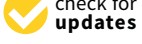



## 1. Introduction

The production of dry-cured hams is traditionally associated with European Mediterranean countries, especially Spain, Italy, France, and Croatia, as the countries of origin of numerous different ham types [1–4]. These products are recognised and highly prized for their delicious odour, flavour, and texture [3]. A dry-cured ham is a cured meat product whose preparation involves dry salting, dehydration, and gradual chemical-enzymatic transformations of fresh pork during long-term ripening (over 12 months). These basic production components are common to all types of dry-cured hams, but it should be emphasised that raw material and some technological and environmental production aspects can differ significantly, resulting in different sensory properties of the final product [4]. The most famous Croatian dry-cured hams are Protected Geographical Indication (PGI)-labelled "Dalmatinski pršut" and Protected Designation of Origin (PDO)-labelled "Istarski

pršut" [5,6], both belonging to the EU geographical indication schemes that protect products coming from a specific region in which traditional production takes place. Regarding the PDO-labelled products, raw ingredients come from the region of origin where the entire production takes place, whereas in the case of the PGIs, at least one of the production stages takes place in the region.

Sensory quality of dry-cured meat products depends on various factors, such as the quality of meat raw materials and other ingredients, processing conditions, and microbial ecology. Microbiota involved in ripening and fermentation processes are diverse and complex, and comprise bacteria, yeasts, and moulds [7]. Moulds overgrow the surface of these products during the production process and are characteristic of the production area [8]. Moulds overgrow prosciuttos within a month and a half, and gradually colonise their entire surface [9]. Surface mould abundance and their subsequent spiderweb-like residua are one of the distinctive characteristics of "Istarski pršut" and an indicator of its proper drying and ripening [5]. The predominant mould genera isolated from the surface of these products are *Penicillium* and *Aspergillus* [9–12], because dry-cured meat products are mostly composed of water, proteins and lipids, and therefore represent a rich substrate that can enhance mould growth. At the same time, however, high salt content, low $a_w$ and low ripening temperatures make the surface of these products almost mould-unfriendly. However, these genera are well-adapted to these xerophilic and halophilic conditions and therefore capable of dominating throughout the ripening period [4,12,13].

Surface mould present on these products is generally appreciated because of its enzymatic activity, such as lipolysis, lipid oxidation, proteolysis, and amino acid degradation, which contribute to the development of a characteristic dry-cured ham flavour. Lipolysis is the first step to free fatty acids' oxidation. Secondary fatty acid reactions result in numerous oxidation products, such as aldehydes, alcohols, ketones and others [8,14,15]. Moulds also contain enzymes that can hydrolyse muscle proteins, contributing to the proteolysis and the release of amino acids [8,16]. The presence of surface moulds on dry-cured meat products has other desirable effects, such as the establishment of surface microclimate able to prevent crust formation and flavour diminishment, in addition to antioxidant activity due to oxygen consumption, and a mycelium barrier effect that reduces the penetration of oxygen and light, thus preserving the colour and taste of the product and delaying its rancidity. Surface moulds can also be protective against pathogenic microorganisms [13,17–19]. Nonetheless, if mould growth is not controlled during the production process, unwanted species can be developed, which can affect the sensory properties of the products and reduce their quality through antibiotic and mycotoxin contamination responsible for consumer allergic reactions, antibiotic resistance, and acute and chronic toxicity [13,20].

Dry-cured hams traditionally produced in households are usually not inoculated with starter cultures and are often produced under relatively uncontrolled conditions. As a result, depending on the production area, uncontrolled moulds can appear on their surface and affect different sensory properties [8]. The main differences between "Dalmatinski pršut" and "Istarski pršut" are the geographical region of origin and some technological aspects of their production, because "Istarski pršut", as opposed to "Dalmatinski pršut", is skinless and unsmoked [5,6]. Although some research on the contribution of certain mould species to lipolysis, proteolysis and volatile compounds of dry-cured meat products has been undertaken, the contribution of fungal population to flavour formation and the role of fungi in big meat pieces, such as dry-cured hams, remain unclear. Of note, regarding dry-cured meat products, the most extensively studied mould species are those that can be used as starter cultures such as *P. nalgiovense*, *P. chrysogenum* and *P. camemberti*, whereas the contribution of other species to sensory properties of these products has been poorly investigated [8,14,16–18,21].

Several studies of sensory properties of Croatian protected products "Istarski pršut" and "Dalmatinski pršut", considered to be among the most important traditional dry-cured meat products produced in different Croatian regions, have been conducted in relation to volatile compounds [4,15,22–24]. However, to the best of our knowledge, to date only

one study identified surface moulds overgrowing these long-ripened dry-cured meat products [12], and the relationship between surface moulds and the sensory properties of these products has not been studied yet. This study aimed to identify surface mould species overgrowing "Dalmatinski pršut" and "Istarski pršut" and the effect of surface mycobiota on the sensory properties of the final products.

## 2. Materials and Methods

### 2.1. Dry-Cured Ham Production and Sampling

For the purposes of this study, samples of "Istarski pršut" (*n* = 15) and "Dalmatinski pršut" (*n* = 20) were retrieved from annual fairs held in Istria (west Croatia) and Dalmatia (south Croatia) during 2019. Samples were taken in the amount of 1.5 to 2 kg and were cut of the caudal part of each ham containing m. semimembranosus, m. semitendinosus and m. biceps femoris.

Dry-cured hams were produced in 2018/2019 using traditional recipes and starter culture-free technologies by different producers located in the two regions according to [4,24,25] with slight modifications. Traditional production is characterised by semi-controlled production conditions, so that differences between manufacturers are possible, but major deviations from product specifications are neither allowed nor were found, given that these meat products are marketed as the products of the Protected Designation of Origin ("Istarski pršut") or the Protected Geographical Indication ("Dalmatinski pršut"). "Dalmatinski pršut" samples were trimmed off without pelvic bones, but with skin and subcutaneous adipose tissue, and "Istarski pršut" samples were trimmed off with pelvic bones, but without skin and subcutaneous adipose tissue. Regarding "Dalmatinski pršut", dry salting made use of coarse and table sea salt, whereas "Istarski pršut" was salted using a combination of coarse and table sea salt supplemented with spices (ground black pepper, laurel and rosemary). On the occasion of the first salting, a total of 30 to 40 g of salt per kg of ham was added, and at the second salting 20 to 30 g of salt per kg was added. The salting took place in cooling chambers at a temperature of 0–5 °C and relative humidity of 80–90% for up to a month, depending on the raw ham weight. After the salting phase, the pressing of hams using approximately 0.1 kg cm$^{-2}$ was applied for 7 days. Drying of both ham types took place in chambers with controlled microclimatic conditions (temperature 12–16 °C; relative humidity 85 to 70%). At the beginning of the "Dalmatinski pršut" drying phase, cold smoke was applied for 20 days at temperatures lower than 22 °C. After 4 months of drying, the hams were moved into ripening chambers operating at the temperatures of 13–18 °C and relative humidity of 65–75%. The production of "Dalmatinski pršut" takes 15 months, and the production of "Istarski pršut" takes 17 months.

### 2.2. Isolation and Traditional Identification of Surface Mycobiota

Immediately after dry-cured hams' delivery, visible mould colonies were removed from (cut off) the surface using a scalpel, followed by the swabbing of the entire sample surface using damp swabs, with particular attention thereby being paid to the slots (unevenness) present on it. The samples were subsequently transferred onto a DG-18 agar (dichloran 18%-glycerol, Merck, Darmstadt, Germany). After a seven-day incubation in darkness at 25 ± 1 °C, individual mould cultures were sub-cultivated on DG-18 agar, malt extract agar (MEA, BD Difco, Franklin Lakes, NY, USA) and Czapek yeast extract agar (CYA, BD Difco, Franklin Lakes, NY, USA) and incubated for seven days at 25 ± 1 °C in darkness to the end of identification using a traditional method based on macro- and micro-morphology and growth characteristics according to [26,27].

### 2.3. Molecular Identification of Surface Mycobiota

The outcome of the traditional mould identification was corroborated using a molecular method. DNA was first extracted from isolated mould colonies using a DNeasy Plant Mini Kit (Qiagen, Hilden, Germany) according to the manufacturer's instructions. Primers specific for beta-tubulin (benA)—The Bt2a forward primer (5′-GGTAACCAAATCGGTGCT

GCTTTC-3′) and the Bt2b reverse primer (5′-ACCCTCAGTGTAGTGACCCTTGGC-3′)—and calmodulin (CaM) loci—the Cmd5 forward primer (5′-CCGAGTACAAGGARGCCTTC-3′) and the Cmd6 reverse primer (5′-CCGATRGAGGTCATRACGTGG-3′)—were selected for polymerase chain reaction (PCR) amplification (Macrogen, Amsterdam, The Netherlands). The reaction mixture (25 μL) was prepared using 1 μL of the template DNA, 12.5 μL of 2× PCR buffer (HotStarTaq Plus MasterMix Kit, Qiagen, Hilden, Germany), 2.5 μL of 10× Coral Load, 0.4 μM of each primer, and nuclease-free water. PCR was performed in a 51149-2 thermal cycler (Prime Thermal Cycler, Staffordshire, UK) under the following cycling conditions: 95 °C for 5 min followed by 40 cycles at 94 °C for 30 s, 56 °C for 30 s, 72 °C for 60 s, concluding with 72 °C for 10 min. PCR products were checked using gel electrophoresis in 1.5%-agarose gel and visualised using UV trans-illumination (UVIDOC, UVITEC, Cambridge, UK). After purification using an ExoSAP-IT PCR clean-up reagent (Affymetrix, Santa Clara, CA, USA), amplicons were sent to a commercial facility for sequencing (Macrogen, Amsterdam, the Netherlands). Sequences were aligned and edited using the DNASTAR Software 16 (Lasergene, Madison, WI, USA), and then compared to those available from the GenBank database using the BLAST algorithm. Obtained sequences were deposited in the GenBank database with accession numbers as follows: OK442622 for *A. proliferans*, OK442623 for *A. pseudoglaucus*, OK562707 for *A. chevalieri*, OK562740 for *A. cibarius*, OK442617 for *P. solitum*, OK442619 for *P. nalgiovense*, OK442616 for *P. citrinum*, OK442620 for *P. salamii*, OK442618 for *P. polonicum*, OK562705 for *P. oxalicum*, OK562706 for *P. bialowiezense* and OK442621 for *A. niger*.

### 2.4. Sensory Evaluation

Sensory evaluation of dry-cured ham samples was conducted by a trained panel of 9 assessors (5 males and 4 females) with an age range of 34 to 60 years (mean age = 49.3, SD = 8.9). The assessors were selected and generically trained according to [28]. In total, 13 training sessions of 60 min each took place. Basic training on how to use the scale and attribute generation based on the previous definitions [29] and attributes generated during training took place in the first four sessions. Nine training sessions were then held, during which panellists were trained on the quantification of the selected terms and calibration across assessors for improved consistency and reproducibility. Sensory analysis was carried out in the sensory laboratory of the Faculty of Food Technology and Biotechnology University of Zagreb according to [30] (room-related technical requirements: relative humidity 50–55%, temperature 20–22 °C) and illumination of 4000 K and 500 lux provided for the working table. Prior to sensory evaluation, assessors gave informed consent. Sensory evaluation made use of a quantitative descriptive analysis (QDA) based on the numerical and unipolar intensity scale developed in collaboration with the Centro Studi Assaggiatori (Brescia, Italy). The intensity of each sensory property was estimated using a numerical scale calibrated from left to right, with "0" indicating the absence of a given sensory property and "9" indicating its strongest intensity. Definitions and ranges of sensory traits are given in Table 1.

Individually coded samples were served at room temperature (two 1.5 mm-thick slices) in sensory booths. Samples were presented in a monadic manner in a randomised order and a 5 min break was taken between samples. In total, seven sessions were held over 4 days, and within each session five samples and one replicated sample were assessed. The replicated sample was used for the calculation of the assessor's repeatability (the calculation was based on the differences in absolute value between the scores assigned to the different descriptors of the sample and its replica) within Big Sensory Soft (Centro Studi Assaggiatori, Brescia, Italy). Water, yogurt, sour apple, and unsalted bread were provided to assessors between samples as palate cleansers. Sensory analysis embraced the assessment of visual qualities (colour of the muscle tissue, colour uniformity, colour of the subcutaneous adipose tissue—only for "Dalmatinski pršut", marbling, surface humidity, tyrosine crystals), olfactory qualities (favourable odour, unfavourable odour, smoky odour—only for "Dalmatinski pršut"), texture (tenderness, juiciness), mouth feel

(saltiness, sweetness, sourness, bitterness) and aroma (specific aroma generated by aromatic herbs or butter, biochemical product properties, fresh meat aroma, moulds).

**Table 1.** Sensory traits, definitions and range implemented for sensory evaluation of dry-cured ham samples.

| Sensory Traits | Definition (Range) |
|---|---|
| **Appearance** | |
| Colour of the muscle tissue | Intensity of red colour in the lean (pale pink to dark red) |
| Colour uniformity | Presence of colour homogeneity in the different muscles of a transversal cut (very low to very high) |
| Colour of the subcutaneous adipose tissue | Intensity of yellow colour of the fat (white to intense yellow) |
| Marbling | Level of visible intramuscular fat (very lean to intense marbled) |
| Surface humidity | Impresion of the surface wetness (very dry to very wet) |
| Tyrosine crystals | Presence of tyrosine crystals on the surface of a transversal cut (none to more than 10 crystals) |
| **Odour** | |
| Favourable odour | Intensity of the typical odour from cured meat products (very low to very high) |
| Unfavourable odour | Intensity of the off-odours (very low to very high) |
| Smoky odour | Perception of any type of smoke aroma (very low to very high) |
| **Texture** | |
| Tenderness | Effort required to bite thorough lean and to convert the sample to a swallowable state (very firm to very tender) |
| Juiciness | Impression of the release of juice during mastication (not to very juicy) |
| **Taste** | |
| Saltiness | Intensity of taste associated with sodium ions (not to very salty) |
| Sweetness | Intensity of sweetness (not to very sweet) |
| Sourness | Intensity of taste associated with citric acid (not to very sour) |
| Bitterness | Intensity of taste associated with caffeine (not to very bitter) |
| **Aroma** | |
| Specific aroma | Intensity of aroma associated with aromatic herbs (Istrian ham) and buttery aroma (Dalmatian ham) (very low to very high) |
| Biochemical aroma | Intensity of aroma associated with oxidised fat (very low to very high) |
| Fresh meat aroma | Intensity of aroma associated with fresh pork meat (very low to very high) |
| Moulds | Intensity of aroma associated with moulds (very low to very high) |

## 2.5. Statistical Analysis

Statistical analyses were performed using the SPSS Statistics Software 22.0 (IBM, New York, NY, USA) and the Big Sensory Soft (Centro Studi Assaggiatori, Brescia, Italy). The results were tested for the normality of their distribution using the Shapiro–Wilk test. In order to determine the statistical significance of differences in sensory and mycological parameters between the two dry-cured hams coming from different regions, the independent samples t-test and Mann–Whitney U test were used. Decisions on statistical relevance were made at the significance level of $p < 0.05$. The results were subjected to principal

component analysis (PCA) to interpret sensory attributes and mycological parameters of the two dry-cured hams using the Statistica 10.0 Software (StatSoft, Palo Alto, CA, USA).

## 3. Results and Discussion

### 3.1. The Presence of Mycobiota on Dry-Cured Ham Surface

The relative prevalence of mould strains growing on a dry-cured ham surface depends on the production area and technological production parameters. The latter include product ripening longevity, regional environmental conditions, temperature, and relative humidity during drying and ripening, and have a significant influence on home-based production during which these parameters are most often uncontrolled. The presence of moulds on the surface of dry-cured meat products is also dependent on physicochemical properties of the product, such as $a_w$, pH and salt content [9,31]. Given that the investigated types of dry-cured hams are protected at the national level in terms of protected geographical indication ("Dalmatinski pršut") and protected designation of origin ("Istarski pršut"), their production technologies are described in detail in Product Specifications [5,6]. The production of both types of dry-cured hams takes at least a year and is characterised by long-term ripening, yielding a finished product water activity ($a_w$) below 0.93 and a mass fraction of salt of 7.5% ("Dalmatinski pršut") and 8% ("Istarski pršut") at the maximum. These dry-cured hams are produced without inoculation of bacteria and mould starter cultures, so that wild moulds spontaneously overgrow their surfaces, despite continuous washing and brushing during the ripening period, intended to prevent an excessive mould growth. Earlier studies have shown that these products have characteristic pH values of less than 6.15 and less than 6.26, $a_w$ below 0.89 and 0.85, and salt content of 6.0–9.2% and 6.0–9.8%, respectively [15,24,32]. Their ripening temperatures range from 12 to 19 °C [32], enabling the growth of species of the *Penicillium* and the *Aspergillus* genera [9,12].

Mould species identified on the surface of "Istarski pršut" and "Dalmatinski pršut" in this study are shown in Table 2.

**Table 2.** Surface mycobiota found on "Istarski pršut" and "Dalmatinski pršut".

| Genus | Species | Number of Isolates | "Istarski Pršut" (*n* = 15) | | "Dalmatinski Pršut" (*n* = 20) | |
|---|---|---|---|---|---|---|
| | | | Dr (%) | Fr (%) | Dr (%) | Fr (%) |
| | *P. salamii* | 6 | 9 | 43 | - | - |
| | *P. polonicum* | 2 | 3 | 14 | - | - |
| | *P. citrinum* | 14 | 9 | 43 | 12 | 40 |
| *Penicillium* | *P. bialowienzense* | 4 | 6 | 29 | - | - |
| | *P. solitum* | 1 | - | - | 2 | 5 |
| | *P. nalgiovense* | 2 | - | - | 3 | 10 |
| | *P. oxalicum* | 2 | - | - | 3 | 10 |
| | *A. proliferans* | 14 | 13 | 64 | 8 | 25 |
| | *A. chevalieri* | 13 | 5 | 21 | 15 | 50 |
| *Aspergillus* | *A. pseudoglaucus* | 1 | - | - | 2 | 5 |
| | *A. niger* | 1 | - | - | 2 | 5 |
| | *A. cibarius* | 7 | - | - | 10 | 35 |

Dr (relative density) = the number of isolates of a given species/total number of isolates × 100. Fr = (relative frequency) the number of samples on which a given species was present/total number of samples × 100.

Examination of all dry-cured hams samples clearly shows that the species of the *Penicillium* and those of the *Aspergillus* genus are equally represented, given that 46% of the identified moulds were of the *Penicillium* and the remaining 54% of the *Aspergillus* genus, with no significant difference in the percentage of isolates between these two genera ($p$ = 0.346). A total of 67 mould isolates and 12 different species (seven *Penicillium* and five *Aspergillus* species) were identified on 35 samples of both dry-cured ham types. The most represented species (listed together with the pertaining relative densities, Dr% = the number of isolates of a given species/total number of fungi isolated × 100) were *A.*

*proliferans* (Dr = 21%), *P. citrinum* (Dr = 21%) and *A. chevalieri* (Dr = 20%). These three species were the only ones found on the surfaces of both "Istarski pršut" and "Dalmatinski pršut". *P. citrinum* is generally taken to be one of most common fungal species on Earth, whereas citrinin conidiospores are one of the most common spores found in the air. Its ability to grow at $a_w$ lower than 0.80 and a temperature of 5 °C helps this species to secure a niche in a wide range of habitats [33]. The highest prevalence of *A. chevalieri* and *A. proliferans* can be explained by the observation that the ascospores of *Aspergillus* teleomorphs can survive a wide range of temperatures (4–43 °C), and very low water activity (0.71) for up to 120 days. Moreover, *A. chevalieri* ranks as one of most common spoilage fungi on Earth, especially in warmer regions [26].

No statistically significant difference in the percentage of mould isolates between the two dry-cured ham types was found ($p = 0.739$) regardless of different production technology (smoked "Dalmatinski pršut"/non-smoked "Istarski pršut") and different climate in their production regions (hot Dalmatia vs. moderate climate Istria). Mycobiota on the surface of these products can be different depending on geographical and climate conditions; as the name itself says, "Istarski pršut" is produced in Istria and "Dalmatinski pršut" in Dalmatia. Dalmatia is a Croatian region known for its extremely hot and dry summers, often comparable to tropical and subtropical areas. In the hottest months, the day temperature frequently rises above 35 °C, whereas even in the coldest months it is maintained above 4 °C. Istria has a moderate Mediterranean climate, with a mean winter temperature in the coldest months of about 6 °C, and a mean summer temperature of 24 °C in the hottest months [5,6].

Similar results were obtained earlier by [12,34], who also failed to find any significant difference in the percentage of mould isolates found on the surfaces of smoked and non-smoked dry-fermented sausages and dry-cured hams coming from different Croatian regions. It was determined that 55% of mould isolates, among which 18% of the *Penicillium* and 37% of the *Aspergillus* genus, populate "Dalmatinski pršut", whereas 45% of isolates, among which 27% of the *Penicillium* and 18% of the *Aspergillus* genus, pertain to "Istarski pršut". A greater variety of *Aspergillus* species was observed with "Dalmatinski pršut" in comparison with "Istarski pršut", as also seen in a study by Zadravec et al. [12]. This was attributed to the climatic environment in which "Dalmatinski pršut" is produced, which is comparable to the subtropical and tropical areas that *Aspergillus* species prefer, contrary to *Penicillium* species, which favour lower environmental temperatures for their growth [26]. Earlier studies reported a decrease in the *Penicillium* population during ripening of dry-cured hams, whereas the *Aspergillus* population was isolated more frequently during post-drying and until the end-ripening phase, probably due to the higher resistance of its spores to drying of the air and ham, and increased temperature during summer ripening. Regarding the general number of mould isolates, at the end of the ripening period this was similar to that in the pre-ripening phase; the same result was found for the number of species [9,10].

"Dalmatinski pršut" showed a greater mould species diversity, with five *Aspergillus* and four *Penicillium* species identified, and the predominance of *A. chevalieri*, *P. citrinum* and *A. cibarius*. On the "Istarski pršut" surface, four *Penicillium* and only two *Aspergillus* species were identified, with the predominance of *A. proliferans*, *P. citrinum* and *P. salamii*. *A. cibarius*, *A. chevalieri* and *A. proliferans* belong to the resistant *Aspergillus* teleomorphs (*Eurotium*-type). *P. salamii* is closely related to the known starter culture *P. nalgiovense*, both species thereby occurring on cured meat products worldwide [35]. *P. salamii*, *P. polonicum* and *P. bialowienzense* were isolated only on "Istarski pršut", whereas *P. solitum*, *P. nalgiovense*, *P. oxalicum*, *A. pseudoglaucus*, *A. niger* and *A. cibarius* were found only on "Dalmatinski pršut". The diversity of species colonising "Istarski pršut" and "Dalmatinski pršut" can also be attributed to the different environment around ripening chambers from which mould spores mostly come, in addition to the ripening conditions in these chambers, because homemade dry-cured hams often ripen uncontrolled [8]. This should also be taken into account when comparing the species identified in this study to the mould species

found to be predominant in other studies of "Dalmatinski pršut" and "Istarski pršut" [9,12] or dry-cured hams from other countries [8,10,31,36].

To the best to our knowledge, only two studies have been conducted on surface mycobiota of these two types of dry-cured hams, as a part of the PDO or the PGI denominations [9,12]. In the research of "Dalmatinski pršut" and "Istarski pršut" by Zadravec et al. [12], a higher number of species (19–20) was identified in comparison with this study (9–12). In both ham types, the dominating species were *P. solitum, P. nalgiovense, P. commune,* and *P. polonicum*. With the exception of *P. commune,* the species mentioned above were also found on ham surfaces in this study, but not as dominating ones. In that research [12], *P. citrinum* was found only occasionally in comparison with this study. Regarding the species of the *Aspergillus* genus, in addition to *A. proliferans, A. pseudoglaucus,* and toxigenic *A. niger*, in the research by Zadravec et al. [12] *A. versicolor* and *A. ochraceus* were also isolated; however, *Aspergillus* sp. were isolated less frequently and not as predominant species, as opposed to the present research. In the study of Istrian ham by Comi et al. [9], the most frequently isolated species were *P. frequentas, P. verrucosum, P. lanosocoeruleum, P. chrysogenum, P. commune* and *P. expansum,* but these species were identified using traditional rather than molecular methods.

In contrast to our research, the dominant mould species found on hams originating from other European countries—for example, *P. commune* and *P. chrysogenum* found on Spanish hams such as the Iberian prosciutto [36,37]—were not identified in the present study. The Italian ham San Daniele was reported to harbour *P. chrysogenum* and *A. fumigatus* [10], whereas the Norwegian hams were found to host *P. nalgiovense, P. solitum* and *P. commune* [31]. Slovenian hams were claimed to harbour *P. nordicum, P. nalgiovense* and *P. milanense* [11]. In general, the higher representation and greater diversity of *Aspergillus* species found in our research can be explained by the fact that Istrian, and especially Dalmatian, prosciuttos ripen in warmer and drier climates compared to the prosciuttos investigated in the studies quoted above. Moulds can influence the sensory properties of dry-cured hams, such as their surface coverage and appearance. Moulds can imbue a favourable white or greyish fungal coverage, which is deemed to be produced by *P. salamii* and *P. nalgiovense*, whereas brown/green sporulation and black spot formation are deemed to be signs of spoilage. The study of sausages inoculated with *P. salamii* and *P. nalgiovense* evidenced a higher favourable contribution of *P. salamii* to sausage coverage, appearance, and flavour, compared to that of *P. nalgiovense* [14,37]. Due to their enzymatic activity, moulds contribute to the nascence of volatile compounds; however, the results of the study by Martín et al. [14] showed that the differences in volatile compounds produced by different fungal populations do not impact the flavour score of the sensory analysis but enhance the overall acceptability due to the improved texture.

It is known that some of the surface moulds can produce mycotoxins as their secondary metabolites, which can have a number of adverse effects on human and animal health [20,35]. In previous research of Croatian dry-fermented meat products, contamination with ochratoxin A (OTA), citrinin (CIT), aflatoxin B1 (AFB$_1$) and cyclopiazonic acid (CPA) was evidenced [20,38–40], consistent with other types of dry-cured meat products originating from other countries [41–44]. The mycotoxigenic mould species identified in this study were *Penicillium citrinum* as a CIT-producer, *Penicillium polonicum* as a verrucosidin and nephrotoxic glycopeptides producer, and *Aspergillus niger* as an OTA producer. "Dalmatinski pršut" and "Istarski pršut" surfaces hosted a similar percentage of mycotoxigenic species (14% and 12%, respectively).

### 3.2. The Effect of Surface Mycobiota on Sensory Properties

The results of the sensory evaluation of "Istarski pršut" and "Dalmatinski pršut" and quantitative descriptive analysis are shown in Figure 1.

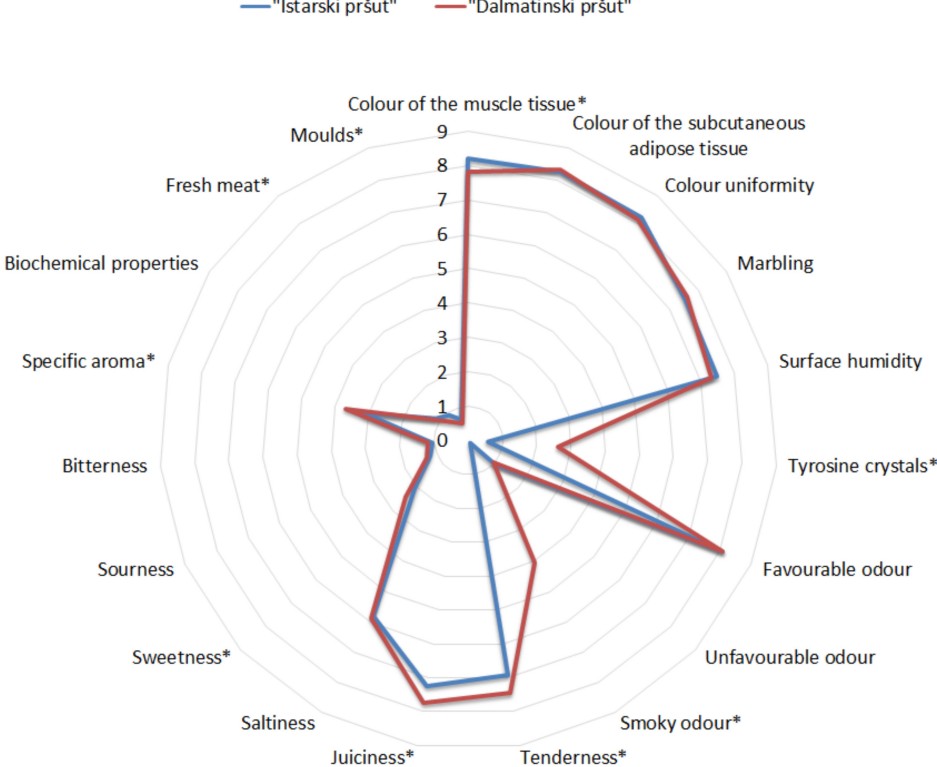

**Figure 1.** Sensory characteristics of "Istarski pršut" and "Dalmatinski pršut" (appearance, odour, texture, taste, and aroma). * statistically significant difference; numerical scale of intensity of sensory properties: "0" indicates the absence of a given sensory property, and "9" indicates its strongest intensity.

Statistical analysis revealed significant differences ($p < 0.05$) between "Istarski pršut" and "Dalmatinski pršut" in 9 of 20 attributes. The differences were found in visual qualities, odour, texture, taste, and aroma, as follows: colour of the muscle tissue ($p < 0.001$), tyrosine crystals ($p < 0.001$), smoky odour ($p < 0.001$), tenderness ($p < 0.001$), juiciness ($p < 0.001$), sweetness ($p = 0.004$), specific aromas ($p = 0.048$), fresh meat aroma ($p = 0.001$) and mould aroma ($p = 0.047$).

In terms of appearance, significant variations were found in the colour of the muscle tissue and tyrosine crystals content, where "Istarski pršut" had a more intense muscle colour and lower tyrosine crystals content. According to the Product Specifications, visual characteristics of "Istarski pršut" and "Dalmatinski pršut" should embrace evenly red to bright red muscle tissue colour; white to pink-white subcutaneous adipose tissue in "Dalmatinski pršut; and a lack of skin and subcutaneous adipose tissue in "Istarski pršut" [5,6]. "Istarski pršut" is allowed to be produced from raw hams obtained from pigs older than 9 months, having a live weight of at least 160 kg. The fulfilment of these requirements results in a higher myoglobin meat content [45] that affects the muscle colour. Furthermore, in dry-cured hams, salt diffusions play an important role in the development of colour, whereas surface moulds can stabilise it. However, diverse factors of other natures can also affect meat colour, such as fermentation time, smoking, and storage [46]. Tyrosine crystals are formed in a ham due to long ripening, which enhances the precipitation of the amino acid tyrosine responsible for white spot appearance, also indicating possible longer maturation and more intense proteolysis [47] that can be enhanced by surface moulds.

Acceptability of dry-cured hams is also highly dependent on texture parameters. According to the Product Specifications, "Dalmatinski pršut" should have a soft chewy consistency, whereas hard consistency and minimal solubility are deemed unacceptable [6]. Tenderness and juiciness of "Dalmatinski pršut" scored significantly higher than those of

"Istarski pršut". The study of Martín et al. [14] showed that the softer texture of inoculated dry-cured hams is a direct consequence of mould inoculation. An early settling of moulds on hams prevents excessive drying of the surface, thus lowering dryness. In addition, proteolysis increased by moulds contributes to lower toughness. Ludemann et al. [48] showed inter- and intra- mould species differences in proteolytic activities, and that the addition of salt stimulates proteolytic activity of mould strains at 25 °C. *P. nalgiovense*, which was found on "Dalmatinski pršut", is known for its proteolytic activity [8,17]. Proteolysis increased by surface moulds may explain the greater tenderness of "Dalmatinski pršut".

The "Istarski pršut" Product Specifications require that the product should have a mild, moderately salty taste, whereas "Dalmatinski pršut" is expected to have a mildly saltish or salty taste; an overly salty or sour/bitter taste, or an entangled and undefinable mixture of tastes, are not allowed [5,6]. Significant variations between the products under study were only found for their sweetness, "Dalmatinski pršut" being sweeter than "Istarski pršut". The overall saltiness of the products was characterised as moderate (5.8–5.9), whereas sweetness, sourness and bitterness achieved low scores (2.1–2.5, 1.2–1.3, and 1.1–1.2, respectively). In the research by Petričević et al. [4], the sweetness of "Istarski pršut" was more pronounced than that of "Dalmatinski pršut". Volatile compounds, such as 2-ethyl furane, 2-pentanone, heptane and octane contribute to the sensory perceptions of sweetness [49]. In the study of "Istarski pršut" by Marušić et al. [23], sweetness and the presence of esters formed by esterification of carboxylic acids and alcohols were positively correlated (the higher the concentrations of esters, the more pronounced the sweetness of the product). Studies have confirmed the ability of *Mucor* and *Penicillium* strains to produce esters [19]. In the study by Bruna et al. [50], the highest levels of esters were detected in superficially inoculated sausages in comparison with uninoculated control sausages, which indicates the contribution of mould metabolic activity to the sweetness of a product.

A salty taste is the result of addition of sodium chloride in the salting stage, whereas a bitter taste is the result of vivid proteolysis, which generates free hydrophobic amino acids and peptides responsible for the taste. A sour taste also originates from amino acids and short free fatty acids produced during proteolysis and lipolysis [51]. Martín et al. [16] showed higher concentrations of some polar amino acids (Asp, Glu, HIs, Thr, Arg and Pro) in inoculated dry-cured hams, as opposed to Iberian hams overgrown by an uncontrolled fungal population. Conversely, some less polar amino acids (Ile, Leu and Trp) were shown to be less present in inoculated hams than in those overgrown by an uncontrolled fungal population. A higher polar/lipophilic free amino acids' ratio is considered flavour-enhancing due to the high correlation between bitterness and lipophilic free amino acids [16]. For the sake of clarity, the comparison with the results of studies devoted to inoculated dry-cured meat products implies meat inoculation with species having beneficial effects, such as *P. chrysogenum*, *P. nalgiovense*, *P. aurantiogriseum*, *P. camemberti*, *P. salamii* and *P. solitum* [13,16,48,50,52].

A smoky odour was reported only for "Dalmatinski pršut", which achieved a score of 4.06. A smoky aroma represents the most distinctive feature of "Dalmatinski pršut" compared to other types of dry-cured hams produced in the broader region; the feature in question makes the product easiest to recognise, whereas the Product Specifications require this odour to be just mild [6]. Phenolic compounds are mainly responsible for the unique aroma and taste of smoked products; in the research of volatile compounds of Croatian dry-cured hams, 18 phenols were identified in smoked dry-cured hams, whereas unsmoked products contained 4-methylphenol only [4].

A specific aroma can be developed from aromatic herbs or butter. It has been shown that terpenes are positively correlated with the flavour coming from added herbal spices, such as pepper and rosemary, which are added in the salting phase of "Istarski pršut" production [23]. For the specific buttery aroma, ketones, such as 2,3 butanedione, are responsible [51]. The "Dalmatinski pršut" Product Specifications require the product to have pleasant aromas of fermented, salted, dry and smoked pig meat, and to be free of any strange odours (such as tar, oil, fresh meat, wet or dry grass odours) [6]. Mouldy and

fresh meat aromas were significantly more pronounced in the case of "Istarski pršut", but these attributes, together with biochemical properties, generally scored very low (0.5–0.6; 0.7–0.9 and <1.1, respectively). A fresh meat aroma is more pronounced with shorter-ripened products, so that the differences in fresh meat aroma score between "Istarski pršut" and "Dalmatinski pršut" can be attributed to the shorter ripening of "Dalmatinski pršut" [23]. In addition, active fungal metabolism of lipophilic amino acids increases the content of volatile compounds related to these amino acids, such as branched aldehydes and carboxylic acids associated with the distinct flavour of dry-cured, long-maturing and aged products. Therefore, the differences in fresh pork meat aroma between "Istarski pršut" and "Dalmatinski pršut" may also be the result of a more vivid metabolic mould activity taking place on the surface of "Dalmatinski pršut", thus generating lipophilic amino acids, and, consequently, the volatile compounds mentioned above [16].

To determine the relationship between sensory traits and surface moulds of the two types of dry-cured ham, PCA analysis was performed, as shown in Figures 2 and 3.

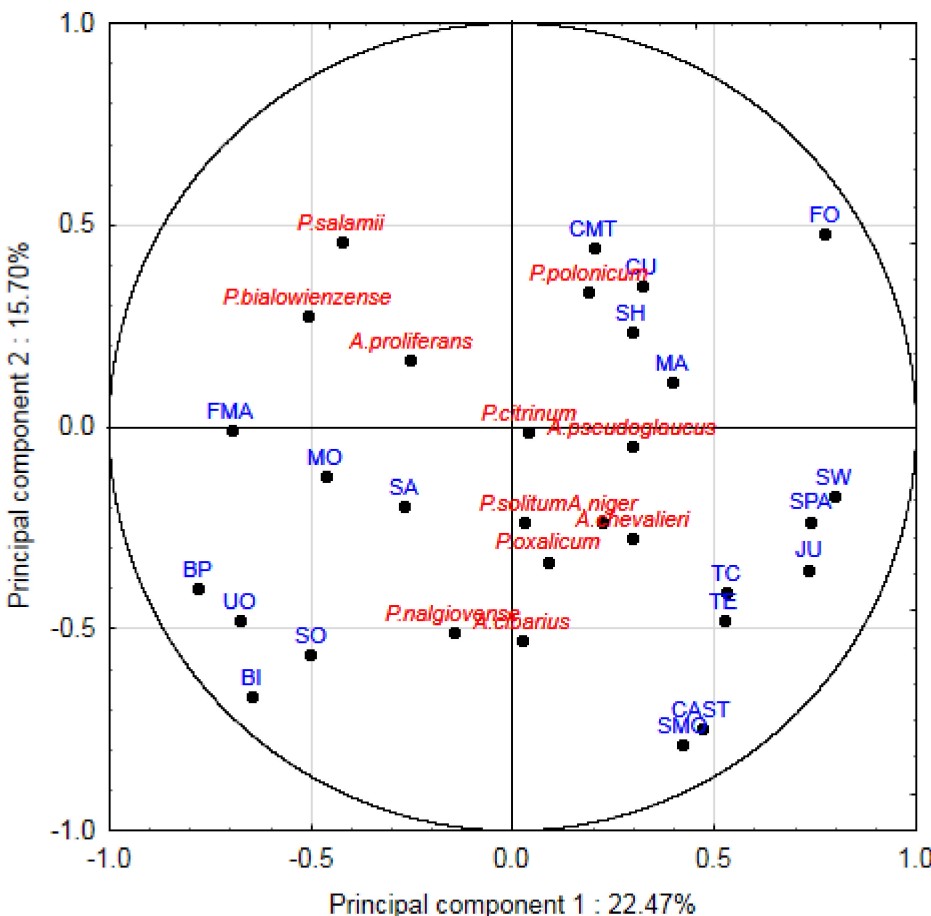

**Figure 2.** Score plot of the variables encompassed by the principal component analysis of sensory attributes and mould species of "Istarski pršut" and "Dalmatinski pršut". CMT = colour of the muscle tissue; CAST = colour of the subcutaneous adipose tissue; CU = colour uniformity; MA = marbling; SH = surface humidity; TC = tyrosine crystals; FO = favourable odour; UO = unfavourable odour; SMO = smoky odour; TE = tenderness; JU = juiciness; SA = saltiness; SW = sweetness; SO = sourness; BI = bitterness; SPA = specific aroma; BP = biochemical properties; FMA = fresh meat aroma; MO = moulds.

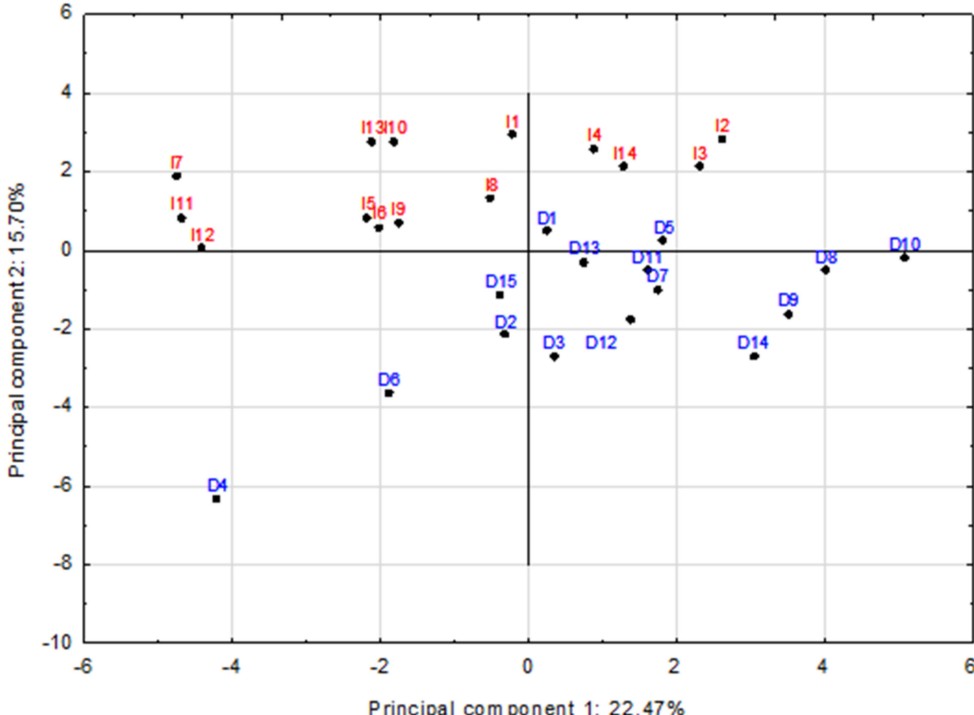

**Figure 3.** Score plot of the principal component analysis of the sensory attributes and mould species of "Istarski pršut" (I) and "Dalmatinski pršut" (D) samples.

The measurements and PCs were interpreted based on the correlations between each parameter and each PC. Thus, variables that were close to each other were considered positively correlated, those separated by 180° as negatively correlated, and those separated by 90° as independently correlated. As can be seen in Figure 2, which shows the relationship between variables, favourable odour, sweetness, specific aroma, and juiciness were the main contributors to the positive side, whereas biochemical aroma and fresh meat aroma, unfavourable odour, and bitterness accounted for the negative side of PC1. This is also an indication of a strong negative correlation between the desired and undesired sensory properties, as established for artisanal smoked dry-cured ham by Kos et al. [53]. Similar results were achieved by Pham et al. [29], who presented sweetness on one side, and bitterness and biochemical aroma on the other, of the same principal component. The same authors reported that saltiness and cured flavour were close to each other and to the bitterness and rancid aroma, indicating a positive correlation between them, but this was not observed in the present research.

The positive side of PC2 was mostly correlated to the colour of the muscle tissue and colour uniformity. In addition to the smoky odour and the colour of the subcutaneous adipose tissue (characteristic only for "Dalmatinski pršut"), the most important variables on the negative side of PC2 were bitterness and sourness, suggesting a strong positive correlation between the two. Although mould aroma, tyrosine crystals and tenderness of the two ham types were significantly different, their correlation with the principal component arrangement was weak, as was the contribution of marbling, saltiness, and surface humidity. Interestingly, mould species were correlated to PC2, for which the closest relationship of *P. polonicum* and *P. salamii* was established on the positive side, whereas the major contributors to the negative side were *P. cibarius* and *P. nalgiovense*. It was found that other mould species were not important for the characterisation of the first two principal components. However, it was found that *P. polonicum* was closely related to the colour of the muscle tissue, colour uniformity and surface humidity, suggesting a high correlation between them and a possible favourable effect of that mould species on the development of the sensory traits in question.

In the score plot on Figure 3, a clear separation between "Istarski pršut" and "Dalmatinski pršut" samples along PC2 can be observed. "Istarski pršut" samples occupy the upper quadrants of the positive and the negative side of PC1, where the colour of the muscle tissue and colour uniformity, in addition to two mould species (*P. salamii* and *P. polonicum*), made the highest contribution. "Istarski pršut" samples occupy the upper quadrants of the positive and the negative side of the PC1, where the colour of the muscle tissue and colour uniformity, in addition to two mould species (*P. salamii* and *P. polonicum*), made the highest contribution. By comparison, "Dalmatinski pršut" samples dominantly occupy the lower quadrants of the positive and the negative side of PC1, where the sensory traits smoky odour, colour of the subcutaneous adipose tissue and bitterness, and the mould species *P. nalgiovense* and *A. cibarius*, were considered important. Furthermore, it was found that *P. salamii*, *P. bialowienzense*, *P. polonicum* and *A. proliferans* were closely related to the "Istarski pršut" samples, whereas *P. nalgiovense*, *A. cibarius*, *P. oxalicum*, *A. niger*, *A. chevalieri* and *P. solitum* were related to the "Dalmatinski pršut" samples. Therefore, it can be concluded that mould species are an important source of information needed for better ham characterisation.

## 4. Conclusions

The study results showed no significant differences in the number of mould isolates between "Istarski pršut" and "Dalmatinski pršut", produced using different production technologies (smoked/non-smoked) and coming from different climatic regions, but did show species diversity, and higher percentage and greater variety of *Aspergillus* species in "Dalmatinski pršut". Different conditions under which dry-cured hams are processed, and consequently different moulds that develop on their surface, resulted in significant variations in 9 of 20 sensory attributes. Predominant species found on the surface of "Dalmatinski pršut" were *A. chevalieri*, *P. citrinum* and *A. cibarius*, whereas those predominating on the surface of "Istarski pršut" were *A. proliferans*, *P. citrinum* and *P. salamii*. A more pronounced tenderness and juiciness, higher tyrosine crystals content, and lesser fresh meat aroma can indicate higher proteolytic activities of mould species populating the surface of "Dalmatinski pršut" in comparison with "Istarski pršut". The results of the PCA analysis indicated that several sensory traits appeared to be important for ham type distinction. Based on the PCA analysis, mould species are primarily correlated with the second principal component, for which the ham types were clearly separated. Furthermore, different mould species were loaded close to each dry-cured ham type, and are thus a significant source of information needed for better ham characterisation.

**Author Contributions:** Conceptualization, T.L. and J.P.; methodology, I.K., N.V. and M.Z.; software, I.K., N.V. and T.L.; validation, I.K. and N.V.; formal analysis, I.K., N.V., I.P., M.Z. and T.L.; investigation, I.K., N.V., I.P., T.L. and M.Z.; resources, D.M. and E.L.; data curation, N.V., I.K. and T.L.; writing—original draft preparation, T.L.; writing—review and editing, J.P.; visualization, T.L. and J.P.; supervision, J.P.; project administration, J.P.; funding acquisition, J.P. All authors have read and agreed to the published version of the manuscript.

**Funding:** This research was funded by the Croatian Science Foundation under the project "Mycotoxins in traditional Croatian meat products: molecular identification of mycotoxin-producing moulds and consumer exposure assessment" (No. IP-2018-01-9017).

**Data Availability Statement:** The sequences of mould isolates included in this study are openly, available in GenBank with accession numbers as described in the Section 2.

**Conflicts of Interest:** The authors declare no conflict of interest.

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
