# Peer review of "The Influence of Surface Mycobiota on Sensory Properties of “Istarski pršut” and “Dalmatinski pršut”"

_processes, doi:10.3390/pr9122287_

Round 1
Reviewer 1 Report
Dear authors,
the manuscript has been revised, I suggest its publication after minor revision. The suggested correction is reported in the attached file.

Author Response
Comment: Are you sure? Usually lactic fermentation is applied to the fermented sausages.
Answer: We agree with our esteemed Referee that the statement in question was incorrect, so that the wording “lactic acid bacteria“has been removed from the pertaining sentence (Line 56).
Comment: Please rewrite the sentence because it is not clearly.
Answer: In line with the Referee’s comment, the unclear sentence addressed above has been rewritten (Lines 97-99).
Comment: Lines 109-135: This information is not usefull in "Materials and methods" section. In my opinion authors have to transfer these concepts to the section of "Results and Discussions".
Answer: We are most obliged to the Referee for this very useful comment. This part of the body text has been removed from the „Material and Methods“ section, and fitted into the “Results and Discussion“ section (Lines 264-274; 308-316).
Comment: Lines 139-144: here more information is needed, i.e. the quantity of salt used (not as reference but the salt actually used), the room temperature, the smoking methodology and the ripening time. Moreover, the sampling method with the samples weight (or dimension) must be reported in this section.
Answer: In line with the Referee’s comment, the technology used in the production of both types of prosciutto under study has been clarified in more detail. For the sake of clarity and precision, the information on the amount of salt used in the production, temperature range at which the production takes place, and the ripening time span has been added, together with details on the Dalmatian prosciutto smoking stage. On top of that, sampling method and the amount of prosciutto retrieved for analytical purposes have been specified, as well (Lines 139-167).
The authors are indebted to our esteemed Reviewer for their most helpful suggestions and comments.
Reviewer 2 Report
In manuscript "The influence of surface mycobiota on sensory properties of
"Istarski pršut" and "Dalmatinski pršut" the influence of mold on sensory properties is described. In the abstract , the first mention of the type of mold should be written with the full name of the mold. The paper analyzes molds from the surface of ham produced in two different areas in Croatia. Have you considered applying molecular techniques directly from the surface of the ham (16S, NGS) without prior cultivation? Namely, the determination of mycobiota would involve the use of molecular methods, not just cultivation.
Have mold isolates tested been tested for potential mycotoxin production? Have ham samples been tested for the presence of mycotoxins?
Author Response
Comment: In manuscript "The influence of surface mycobiota on sensory properties of "Istarski pršut" and "Dalmatinski pršut" the influence of mold on sensory properties is described. In the abstract, the first mention of the type of mold should be written with the full name of the mold.
Answer: In the revised Abstract, full names of the moulds have been specified at the first mention (Lines 24, 25, 29).
Comment: The paper analyzes molds from the surface of ham produced in two different areas in Croatia. Have you considered applying molecular techniques directly from the surface of the ham (16S, NGS) without prior cultivation? Namely, the determination of mycobiota would involve the use of molecular methods, not just cultivation.
Answer: The NGS technique was not used as a molecular method, since the study aimed at identifying individual mould species present on the prosciutto surfaces. The latter was accomplished via multiplication and CaM & beta globulin region sequencing. The identified moulds were linked to the prosciutto sensory properties, so as to ascertain either favourable or unfavourable impact of individual mould species on sensory profile of the studied products. We are grateful to the Referee for this very useful comment and shall strive to apply the NGS technique in our future research in order cover the entire mycobiota growing on the surfaces of these products.
Comment: Have mold isolates tested been tested for potential mycotoxin production? Have ham samples been tested for the presence of mycotoxins?
Answer: Mould species identified within this study frame were not tested for potential mycotoxin production, nor was the presence of mycotoxins verified. This study aimed at establishing the impact of surface moulds on prosciutto sensory properties, while the identification of moulds that might pose as mycotoxin producers was the subject-matter of the previously published contribution of Zadravec et al. (2020) A study of surface moulds and mycotoxins in Croatian traditional dry-cured meat products. The contribution in reference dealt with a number of traditional meat products, prosciuttos being among them. Therefore, the aim of this study was narrowed down to the impact of surface moulds on prosciutto sensory properties, which has not been investigated earlier.
The authors are indebted to our esteemed Reviewer for their most helpful suggestions and comments.